# Unsupervised Continual Learning and Self-Taught Associative Memory Hierarchies*

**James Smith, Seth Baer, Zsolt Kira & Constantine Dovrolis**
College of Computing, Georgia Institute of Technology, Atlanta, GA
{jamessealesmith, sbaer8, zkira, constantine}@gatech.edu

## Abstract

We first pose the Unsupervised Continual Learning (UCL) problem: learning salient representations from a non-stationary stream of unlabeled data in which the number of object classes varies with time. Given limited labeled data just before inference, those representations can also be associated with specific object types to perform classification. To solve the UCL problem, we propose an architecture that involves a single module, called Self-Taught Associative Memory (STAM), which loosely models the function of a cortical column in tha mammalian brain. Hierarchies of STAM modules learn based on a combination of Hebbian learning, online clustering, detection of novel patterns and forgetting outliers, and top-down predictions. We illustrate the operation of STAMs in the context of learning hand-written digits in a continual manner with only 3-12 labeled examples per class. STAMs suggest a promising direction to solve the UCL problem without catastrophic forgetting.

**Introduction.** Unsupervised Continual Learning (UCL) involves learning from a stream of unlabeled data in which the data distribution or number/type of object classes vary with time. UCL is motivated by recent advances in Continual Learning (CL) (Hsu *et al.*, 2018; Parisi *et al.*, 2018) but also differs in that it is completely unsupervised and there are no priors on the data stream.

In UCL the data stream includes unlabeled instances of both previously learned classes and, occasionally, new classes. This setting mirrors the natural world where known object types keep re-appearing – if they do not, it makes sense to forget them. Many CL methods involve some sort of "replay" – we argue that observing instances of known classes (perhaps infrequently) is equivalent to replaying previous instances.

To evaluate whether a given architecture can solve the UCL problem, we partition the time axis in distinct *learning phases*. During each phase, the data stream includes unlabeled examples from a constant set of classes (unknown to the architecture). At the end of each phase, we evaluate the architecture with a simple classification task. To do so, we provide a limited number of labeled instances per class. This labeled dataset is *not* available during the learning phase and it is only used to associate the class-agnostic representations that the architecture has learned with the specific classes that are present in the labeled dataset. This is different than Semi-Supervised Learning (SSL) methods (Springenberg, 2015; Oliver *et al.*, 2018; Miyato *et al.*, 2018) because SSL requires both labeled and unlabeled data during the training process. We have found one SSL method compatible with the UCL problem, the latent-feature discriminate model (M1) (Kingma *et al.*, 2014), and we present a variation of that method in the experimental section.

To solve the UCL problem, we have developed a neuro-inspired architecture based on a model of cortical-columns, referred to as Self-Taught Associative Memory (STAM). The connection between STAMs and cortical models is described in another reference. Due to space constraints, we only present the STAM architecture from a functional perspective here. The architecture is a layered hierarchy of STAM modules that involve forward, feedback, and lateral connections. The hierarchy learns salient features through online clustering. Each feature is a cluster centroid. STAMs at different layers of the hierarchy learn centroids at different spatial resolutions (different receptive

---

*Supported by the Lifelong Learning Machines (L2M) program of DARPA/MTO: Cooperative Agreement HR0011-18-2-0019

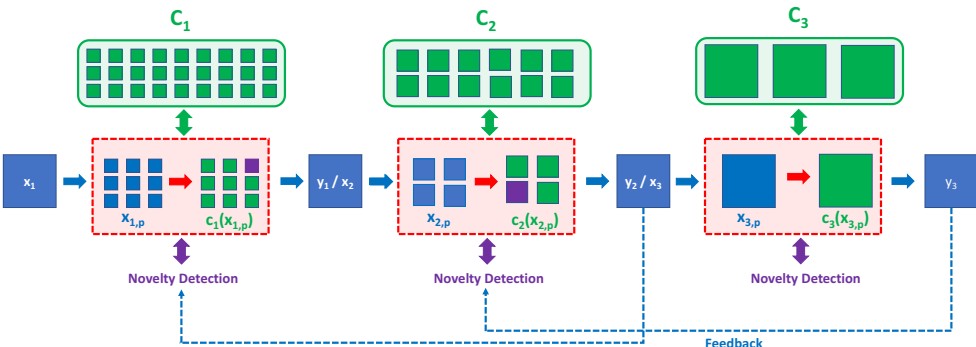

**Figure 1:** Illustration of a 3-layer STAM hierarchy. At each layer, the input is broken up into several overlapping patches (Receptive Fields), which are clustered using online k-means (Euclidean distance). If an input RF is flagged to be novel, a new cluster is created and its centroid is initialized based on that patch. Each layer reconstructs its output image based on the selected centroid for each RF, and that image becomes the input to the next layer. Feedback connections are used to control the creation of new centroids based on higher-layer predictions over wider RFs.

field sizes). STAMs learn in an online manner through mechanisms that include novelty detection, forgetting outlier patterns, intrinsic dimensionality reduction, and top-down predictions.

STAMs have some superficial similarities with Convolutional Neural Networks (CNN) (Krizhevsky *et al.*, 2012) in that they are both layered and have increasing receptive field sizes. However, the STAM architecture learns in a Hebbian manner without the task-specific optimization requirement of CNNs. Further, the features learned by STAMs are highly interpretable (they are basically common patterns at different spatial resolutions), and they adapt to non-stationarities in the data distribution.

The STAM architecture also differs from previous hierarchical clustering schemes such as (Coates *et al.*, 2011; Coates and Ng, 2012) in that STAMs rely on online clustering (to support continual learning), novelty detection (to detect new classes), limited memory (to forget outlier centroids), and intrinsic dimensionality reduction (to generalize across instances of the same class). In general, we do not consider iterative algorithms, such as the "deep clustering" architecture (Caron *et al.*, 2018), to be compatible with UCL because they require repetitive training epochs through the same data.

**Self-Taught Associative Memory (STAM) architecture.** A STAM architecture (illustrated in Figure 1) is composed of $L$ layers. The functional unit at each layer is a STAM module. Layer $i$ consists of $M_i$ STAM modules. In the context of object recognition in static images, each STAM processes a Receptive Field (RF) of the input image in that layer. The RF size gradually increases along the hierarchy (i.e., $M_i$ gradually decreases).

The feedforward input to the $m$'th STAM module of layer $i$ at time $t$ is denoted by $x_{i,m}(t)$. The set $C_i(t)$ of clusters at layer $i$ is shared among all STAMs of that layer. The $j$'th centroid of layer $i$ is denoted by the vector $w_{i,j}(t)$. We drop the time variable $t$ when it is not necessary. Given the set of $C_i$ clusters, each STAM module of layer $i$ selects the nearest centroid to its input based on Euclidean distance:

$$c(x_{i,m}) = \arg \min_{j=1...|C_i|} ||x_{i,m} - w_{i,j}|| \tag{1}$$

The input of layer $i+1$ is the output of the previous layer. The output of layer $i$, denoted by $Y_i$, is of the same (extrinsic) dimensionality with the input $X_i$ in that layer. $Y_i$ is constructed by the sequence of selected centroids, first replacing the input RF $x_{i,m}$ with the corresponding centroid $c(x_{i,m})$, and averaging the overlapping segments. Consequently, the *intrinsic dimensionality* of $Y_i$ is much lower than that of $X_i$: $Y_i$ can take $|C_i|^{M_i}$ distinct values, and $M_i$ decreases along the hierarchy as the RFs get larger.

A STAM learns in an online manner by updating the centroid that has been selected by its input vector. If the $m$'th STAM module selected centroid $j$ at layer $i$ for its input vector $x_{i,m}$, we update

that centroid as follows:

$$w_{i,j} = \alpha\, x_{i,m} + (1 - \alpha)\, w_{i,j}, \text{when } c(x_{i,m}) = j \tag{2}$$

where the constant $\alpha$ is a learning rate parameter $0 < \alpha < 1$. The higher the $\alpha$, the faster the learning process becomes, potentially resulting in lower accuracy. In the rest of this paper, $\alpha$=0.05.

Centroids are created and initialized dynamically, based on a novelty detection algorithm. To detect novel patterns, we estimate in an online manner the mean distance $\mu_j$ and standard difference $\hat{\sigma}_j$ between a centroid $j$ and its assigned inputs:

$$\mu_j = \alpha\, ||x_{i,m} - w_{i,j}|| + (1 - \alpha)\, \mu_j \tag{3}$$

$$\hat{\sigma}_j = \alpha\, |\, ||x_{i,m} - w_{i,j}|| - \mu_j\, | + (1 - \alpha)\, \hat{\sigma}_j \tag{4}$$

Based on the previous two online estimates, an input $x_{i,m}$ is flagged as "novel" if its distance from the nearest centroid $j$ is significantly larger than the centroid's mean distance $\mu_j$ estimate,

$$||x_{i,m} - w_{i,j}|| > \mu_j + 3\,\hat{\sigma}_j \tag{5}$$

If $x_{i,m}$ is flagged as novel, a new centroid is created at layer $i$ and it is initialized based on that input. The number of centroids learned at each layer is fixed: layer $i$ cannot remember more than $C_i$ centroids. When that number is exceeded, the centroid that has been *Least Recently Used* (LRU) is forgotten.

To help differentiate between patterns that are outliers and true novelties, we leverage top-down connections. Suppose that $y_{i+1,m}$ is the portion of $Y_{i+1}$ that corresponds to the $m$'th RF at the $i$th layer, and let $c_i(y_{i+1,m})$ be the layer $i$ centroid that is nearest to to $y_{i+1,m}$. This centroid represents the prediction of layer $i + 1$ for the $m$'th RF at layer $i$. If the corresponding input $x_{i,m}$ at layer $i$ was flagged as novel but $y_{i+1,m}$ does not pass the "novelty" criterion of equation 5, then we do not create a new centroid for that input at layer $i$.

**Classification.**   In principle, we can use any classifier to evaluate the representations (centroids) that the STAM architecture has learned at the end of a learning phase. Here, we describe a simple classifier that first associates each output-layer centroid with a class by calculating the "allegiance" of each labeled input vector $x_n$ to centroid $w_j$ relative to the nearest-neighbor centroid:

$$s_{w_j,x_n} = \frac{e^{-||w_j - x_n||}}{\max_{j'} e^{-||w_{j'} - x_n||}} \tag{6}$$

The allegiance of centroid $w_j$ to class $m$ is simply the average $s_{w_j,x_n}$ across all labeled inputs $x_n$ that belong to class $m$:

$$S_{w_j,m} = \frac{1}{N_m} \sum_{n:y_n=m} s_{w_j,x_n} \tag{7}$$

where $N_m$ is the number of labeled examples of class $m$, and $y_n$ is the class of input $x_n$. It is possible that a centroid at the output layer does not have strong allegiance to any class. For this reason, we remove centroids for which the maximum allegiance $\max_m(S_{w_j,m})$ is less than 70%.

The classification of an input $x$ is based on the distance between $x$ and each centroid as well as the allegiance of each centroid to every class. Specifically, $x$ is assigned to the class $m$ that maximizes the following sum across all centroids $w_j$,

$$k = \arg\max_m \sum_{w_j} S_{w_j,m} e^{-||w_j - x||} \tag{8}$$

**Experiments.**   We divide the time axis into five learning phases. In each learning phase, the data stream includes two additional classes (digits) from the MNIST dataset, i.e., the first learning phase includes only 0s and 1s, while the fifth learning phase includes all ten digits. In each learning phase, the architecture has access to $N_{old}$ unlabeled examples per class of previously learned classes and $N_{new}$ unlabeled examples per class of newly introduced classes. At the end of each phase, we introduce a limited amount of labeled data per class to evaluate classification accuracy.

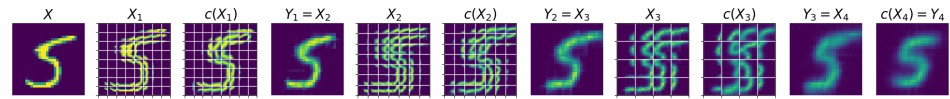

**Figure 2:** An image is processed by a 4-layer STAM hierarchy.

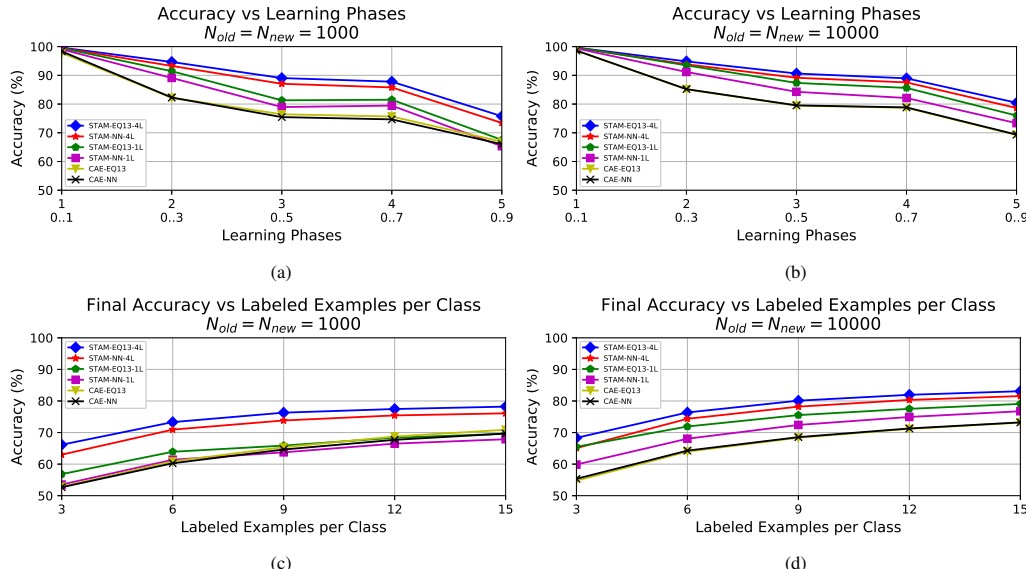

**Figure 3:** (a) UCL results for $N_{old} = N_{new} = 1,000$ and 10 labeled examples per class (b) UCL results for $N_{old} = N_{new} = 10,000$ and 10 labeled examples per class (c) Final phase accuracy vs. labeled examples per class for $N_{old} = N_{new} = 1,000$ (d) Final phase accuracy vs. labeled examples per class for $N_{old} = N_{new} = 10,000$

Together with the STAM architecture, we also train a Convolutional AutoEncoder (CAE) (Li *et al.*, 2017) in an unsupervised manner, and then create a classifier using latent representations of the labeled data for each evaluation period. The CAE architecture was designed specifically for the MNIST dataset, using three convolution and max pooling layers in the encoder and three convolution and upsampling layers in the decoder. We optimize binary cross-entropy loss using the Adam method (Kingma and Ba, 2014). As another baseline, we simply consider a single-layer STAM, which can be interpreted as a non-hierarchical version of the STAM architecture.

For both STAMs and the CAE network, we use two classifiers: nearest-neighbor (NN) and the classifier of (equation 8) – referred to as *EQ8* in the results. We apply EQ8 on the CAE latent representations as centroids with allegiance only to the class corresponding to the input instance's label. The STAM architecture is described in Table 1. We present results (Figures 2-3) for two experiments on the MNIST dataset (Lecun *et al.*, 1998) based on 10 trials, evaluating accuracy on 10,000 images that were not seen during training.

For the first experiment, we compare classification accuracy for various amounts of unlabeled data. We consider $N_{old} = N_{new} = \{1,000, 10,000\}$ and provide 10 labeled examples per class for classification. We observe that the performance of the CAE and single-layer baselines strongly fall off when reducing the unlabeled data to $1,000$, whereas the STAM architecture shows less catastrophic forgetting. For the second experiment, we repeat the first experiment varying the amount of labeled data per class and we report only the classification accuracy at the last learning phase. As expected, STAMs and CAE both see large benefits from increasing the number of labeled examples per class. However, we see that STAM can perform reasonably well with fewer labeled examples compared to the CAE baseline.

**Table 1:** STAM Hierarchy

| Layer | RF size | stride | $|C_i|$ |
|-------|---------|--------|---------|
| 1 | 7 | 3 | 200 |
| 2 | 10 | 3 | 200 |
| 3 | 13 | 5 | 300 |
| 4 | 28 | 28 | 400 |

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
