# OpenReview forum: "Unsupervised Continual Learning and Self-Taught Associative Memory Hierarchies"
_ICLR.cc/2019/Workshop/LLD — LLD 2019_

### Official Review · AnonReviewer2 · 2019-04-07
**Compelling method; some additional analysis would be helpful**

**Rating:** 4
**Confidence:** 2

**Review:**


Paper Summary

This paper proposes using hierarchies of Self-Taught Associative Memory (STAM) modules to solve the Unsupervised Continual Learning  (UCL) problem, wherein salient representations must be learned from a stream of unlabeled data that can be used for classification with a small amount of labeled data provided at a later time.  Importantly, the stream is assumed to be non-stationary in the sense that the number of classes in the stream varies with time and carries no associated prior that is known to the modeler.  The paper describes the STAM approach and presents compelling evidence that the representations it learns are well-suited for few-shot classification when compared with reasonable baselines.


Quality (Pros)
(1) The UCL problem is clearly described

(2) The STAM architecture represents an interesting method for learning a representation that takes advantage of hierarchical receptive fields in a similar manner to CNNs, but is adaptable to changing data distributions by design via the online clustering and outlier pruning steps.

(3) The association of learned representations with different classes is accomplished in a reasonable way

(4) Experimental results suggest that the STAM method consistently outperforms the CAE baseline


Limitations (Cons) and Questions
(1) Additional motivation for the UCL problem would be welcome, as well as experiments on additional datasets that would demonstrate a wider variety of use cases

(2) It is unclear how such hyperparameters as the number of standard differences used in the novelty detector, the number of clusters chosen at each level in the hierarchy, and the allegiance value at which centroids are removed at the classification stage affect performance.  How much computational effort is required to find these values?  And how much does performance depend on them?

(3) Computational efficiency is not touched upon; from a systems standpoint, what is the cost of this model relative to the CAE baseline?

(4) I would suggest adding N = 1,000, 10,000 numbers to Figure 3 -- it is currently hard to read and contextualize.  It would also help to show the difference between 1,00 and 10,000 on a single graph, perhaps, as it is difficult do see the relative changes as currently presented

(5) A point that I found confusing was exactly how the outliers are detected in the second paragraph under equation (5).  Specifically, from my reading, I was under the impression that y_{i+1,m} would be equivalent to c_i(y_{i+1,m}) except for any differences caused by averaging with other overlapping patches.  Is this averaging with overlapping patches then the reason that y_{i+1,m} and c_i(y_{i+1,m}) could be different?  Or have I misunderstood?  Regardless, some clarifying language around this point could be helpful to the reader.


Clarity

The presentation is generally clear and well-written, modulo the above suggestions.


Significance

The STAM approach seems to be a compelling method to solve the UCL problem based on the analysis presented, and comparison to baselines seems reasonable.  This method could be useful in a number of machine learning applications.

---

### Decision · Program_Chairs · 2019-04-16
**Acceptance Decision**

Accept